# Textual Feature Extraction Using Ant Colony Optimization for Hate Speech Classification

**Shilpa Gite** [1,*], **Shruti Patil** [1,*], **Deepak Dharrao** [2,*], **Madhuri Yadav** [1], **Sneha Basak** [1], **Arundarasi Rajendran** [1] **and Ketan Kotecha** [3]

[1] Symbiosis Centre for Applied Artificial Intelligence, Department of Artificial Intelligence and Machine Learning, Symbiosis Institute of Technology, Symbiosis International (Deemed University), Pune 412115, India
[2] Department of Computer Science and Engineering, Symbiosis Institute of Technology, Symbiosis International (Deemed University), Pune 412115, India
[3] Symbiosis Centre for Applied Artificial Intelligence, Symbiosis Institute of Technology, Symbiosis International (Deemed University), Pune 412115, India
[*] Correspondence: shilpa.gite@sitpune.edu.in (S.G.); shruti.patil@sitpune.edu.in (S.P.); deepak.dharrao@sitpune.edu.in (D.D.)

**Abstract:** Feature selection and feature extraction have always been of utmost importance owing to their capability to remove redundant and irrelevant features, reduce the vector space size, control the computational time, and improve performance for more accurate classification tasks, especially in text categorization. These feature engineering techniques can further be optimized using optimization algorithms. This paper proposes a similar framework by implementing one such optimization algorithm, Ant Colony Optimization (ACO), incorporating different feature selection and feature extraction techniques on textual and numerical datasets using four machine learning (ML) models: Logistic Regression (LR), K-Nearest Neighbor (KNN), Stochastic Gradient Descent (SGD), and Random Forest (RF). The aim is to show the difference in the results achieved on both datasets with the help of comparative analysis. The proposed feature selection and feature extraction techniques assist in enhancing the performance of the machine learning model. This research article considers numerical and text-based datasets for stroke prediction and detecting hate speech, respectively. The text dataset is prepared by extracting tweets consisting of positive, negative, and neutral sentiments from Twitter API. A maximum improvement in accuracy of 10.07% is observed for Random Forest with the TF-IDF feature extraction technique on the application of ACO. Besides, this study also highlights the limitations of text data that inhibit the performance of machine learning models, justifying the difference of almost 18.43% in accuracy compared to that of numerical data.

**Keywords:** feature engineering; Term Frequency–Inverse Document Frequency (TF-IDF); Bag of Words (BoW); Chi-square test; Ant Colony Optimization (ACO); machine learning

## 1. Introduction

Today, as we look around, we see a global and exponential world in terms of resources and data that needs to be tackled and handled well. It is all about innovation at warp speed when we use these exponential technologies and resources to tackle enormous goals and use quick iteration and fast feedback to accelerate progress toward those goals. One of these goals brings machine learning into the picture, which discerns vast amounts of data every day and tries to maintain sync between the algorithms and the size of the data. Different factors are considered while developing an ML model, one of which is the number of input characteristics affecting the system's performance significantly [1,2]. The amount of information being produced today compounded with the input features poses a severe and exciting challenge for the researchers since more data leading to more features negatively impacts the performance of the traditional methods [3,4].

Hence, the quantity of data in today's datasets necessitates the preprocessing of data and the development of intelligent algorithms for extracting useful information to segregate essential features from non-essential ones [5,6]. For using machine learning methods constructively, preprocessing the data is as essential as modeling the data. Preprocessing of these datasets is frequently done [6,7] for two reasons: (1) to cut down the size of the dataset to achieve more efficient analysis, and (2) to tailor the dataset to best suit the chosen analysis method. These contributions of data preprocessing help to select features, which improves the productivity of ML algorithms. Hence, as the number and variety of datasets grow, it is more critical to reduce them methodically [8].

A feature is nothing but a unique, measurable property of the observed process. Any ML algorithm may do classification using a feature subset. The domain of features utilized in machine learning and pattern recognition applications has grown from tens to hundreds of variables or features in recent years. Several strategies have been developed to solve the challenge of minimizing irrelevant and superfluous variables that bog down strenuous activities. On the contrary, feature selection removes redundant, irrelevant, and inconsistent features and uses the most contributing features in the model training [9]. It aids in many tasks, such as the interpretation of data, the reduction of computation requirements, the reduction of the dimensionality curse, and the improvement of predictor performance. Hence, it is considered one of the primary techniques of preprocessing data and has evolved as an essential constituent of the machine learning process [8].

Redundant, irrelevant, noisy, and inconsistent features can disrupt an algorithm's computational cost, learning performance, and accuracy. Various feature selection methods based explicitly on optimization algorithms can prevent this and ultimately improve the execution of the whole process. Population-based optimization algorithms such as Ant Colony Optimization (ACO) have gained a lot of interest among the various methods developed for selecting features. Such strategies improve results by utilizing what they have learned from past iterations. ACO, inspired by real-life observations of ants searching for the shortest routes to food sources, executes smoothly with low computational complexity and solves one of the significant high-dimensional feature space problems in text categorization [10]. In this experiment, we focus on ACO (a metaheuristic approach to feature selection) to optimize the process of selecting features and understand the challenges faced during the feature selection on textual data in a comparative study. There are other essential optimization techniques inherently incorporated into neural networks. Amelio et al. in [1,3] have discussed the benefits of multilayer network-based approaches.

Most research papers seem to have focused on a concrete methodology revolving around preprocessing techniques or feature selection methods based on optimization algorithms [11] solely on numerical data in detail. Although these papers [12,13] provide a great deal of analysis of different methodologies, they often fail to provide a more flexible approach to bringing both aspects of data preprocessing to the table, be it any generalized techniques or feature selection methods. In contrast, this paper intends to bridge the gap by giving a detailed comparative analysis of the performance achieved by four different machine learning models when employed with and without an optimization algorithm.

In this research work, we required both textually labeled datasets and numerically labeled datasets for our research purpose. Hence, we analyzed two datasets, hate speech and stroke prediction. Sentiment analysis is a subfield of Natural Language Processing (NLP) that involves the automatic identification and categorization of subjective information in text data, typically expressed as positive, negative, or neutral [2]. We chose to use Twitter as a source for extracting our text dataset for sentiment analysis. On the other hand, hate speech is any form of expression that attacks or insults a person or a group based on their race, religion, ethnicity, sexual orientation, or other identities. It is considered harmful and unacceptable and is not protected by freedom of speech laws in many countries [4]. This work involves analyzing the text data to determine the emotional tone or opinion expressed by the writer.

The fundamental objective of this research has also been narrowing down and validating the challenges and limitations faced by text data while working with optimization algorithms by implementing the proposed pipeline on both numerical and textual datasets. It is important to note that sentiment analysis and hate speech detection are complex tasks and require sophisticated models to accurately identify the sentiment or intention behind the words used. It is also important to approach these tasks with cultural sensitivity, as the interpretation of certain words or phrases can vary greatly across different cultures and regions.

*Motivation*

There are numerous methods to solve feature selection problems, broadly categorized into the wrapper, filter, and hybrid methods. Apart from these baseline methods, a unique feature selection technique has been going on for the past few years, called the metaheuristic approach. ACO is one such algorithm that uses a metaheuristic approach to select features. However, its use has been limited to the study of numerical data only. Many researchers have performed different experiments related to the algorithm, with very few being performed on textual datasets. Although the limitations of text data pose a huge hurdle in terms of computational speed and cost, we wanted to study the behavior and performance of such optimization algorithms on text data. Hence, this study endeavors to realize the complexity of the text data and its impact on the performing models when experimented on with optimization algorithms. We have also examined the performance of numerical data against textual data to provide a broader view concerning the limitation of selecting features on text data. The authors' investigations are summarized as follows:

(1) Implement the framework using Ant Colony Optimization (ACO) for feature selection to improve the AI models.
(2) Evaluate these selected features on textual and numerical datasets.
(3) Evaluate and analyze the Logistic Regression (LR), K-Nearest Neighbor (KNN), Stochastic Gradient Descent (SGD), and Random Forest (RF) machine learning algorithms.

The rest of the paper is divided into five sections. The Section 2 outlines the existing work on feature selection and optimization methods. In the Section 3, we discuss the analysis of the datasets and algorithms used. Next is the system design, which presents the meat of the paper, describing the project pipeline, dataset curation, exploratory data analysis, and the models implemented. Lastly, the paper winds up its review by discussing the findings, drawing a conclusion, and presenting the future scope of this study.

## 2. Literature Review

This section shows an overview of the relevant studies on feature selection using the Ant Colony Optimization algorithm. The authors have studied various articles that have used the Ant Colony Optimization algorithm for feature selection. The authors of [14,15] have covered Ant Colony Optimization algorithms for feature selection in a very broad fashion in the literature Table 1.

An ACO-based pipeline to improve text categorization performance is proposed in [10]. Aghdam et al. in [10], performed a comparative analysis with other genetic algorithms to test the proposed system proving that the suggested system outperforms the genetic algorithms. However, this method fails to compare the results with any numerical-based dataset and does not cover the limitation of text features while performing the experiment.

Saraç and Özel in [5] used the ACO algorithm to select the best feature using classifiers C4.5, naïve Bayes, and KNN, showing that the presented algorithm performs better than popular techniques such as Chi-square and information gain. In another work, the novel approach to the ACO algorithm with two fuzzy controllers is used, which adjusts the parameters of ACO adaptively and performs a comparison with respect to genetic algorithms on 10 different datasets. However, none of them contain textual data, which can be seen as a possible limitation [16]. The Multi-Objective Function-Based Ant Colony

Optimization (MONACO) procedure for opinion classification along with popular feature extraction and feature selection techniques achieved a good performance score on the proposed algorithm, but multi-class classification was not taken into account while performing the experiment [17]. The Ant Colony Optimization algorithm shows improved performance by using the Bayesian classification method. The comparative analysis with the PSO algorithm proves that the suggested method performs feature selection efficiently with good classification performance [18,19]. To select minimum features with maximum information, Sabeena S and Balakrishnan Sarojini [20] proposed a feature selection method using ACO for classification tasks but failed to shed light on the performance with text data [20]. This research work by Imani, M. et al. [21] proposed an embedded approach to feature selection using the Chi-square approach for filtering the features. The proposed architecture suggests a hybrid approach with a genetic algorithm and Ant Colony Optimization algorithm as a wrapper method. A comparison with other methods was performed to prove the superiority of the proposed approach. However, the paper does not discuss the effectiveness of the approach on another textual dataset as well as its impact on the data having numerical columns [21].

In another study, the researchers provided hybridization models of Ant Colony Optimization with SVM-ACO, or ACO with a genetic algorithm for selecting more relevant features from multiple datasets to prove the system's superiority [22,23]; however, the experiment has not been tested on the multi-class classification of text data and fails to provide a comparison with standard feature selection methods. Another hybrid strategy for feature subset selection is by connecting the classifier ensemble to the ACO algorithm. However, the approach has not been tested on text data and lacks comparison with standard feature selection methods [24]. The FACO algorithm was developed using the combination of feature selection of network data and Ant Colony Optimization algorithm to improve classification accuracy [25] as presented in Table 1.

**Table 1.** Relevant research papers studied in the literature review from different perspectives.

| Ref. | Summary | Feature Engineering Techniques Used | Performance Measures | Limitations |
|---|---|---|---|---|
| [10] | Proposes an ACO-based pipeline to improve text categorization performance. | Bag of Words, Information gain, Chi-square | 89.08 Micro-F1 and 78.42 Macro-F1 | - Fails to compare the results with any numerical-based dataset.<br>- Does not cover the limitation of text features while performing the experiment. |
| [5] | Used the ACO algorithm to select the best feature using classifiers C4.5, naïve Bayes, and KNN. | Tagged terms, Bag of Terms, Title Tags, URLs | 0.98 F-measure | - Do not discuss the impact of optimization algorithms on multi-class classification. |
| [16] | Introduced a novel approach to the ACO algorithm with two fuzzy controllers. | Information gain | 98.8% Accuracy | - Does provide a comparative analysis of ten benchmark datasets; However, none of the datasets contain textual data, which is a possible limitation. |
| [17] | Proposes a Multi-Objective Function-Based Ant Colony Optimization (MONACO) procedure for opinion classification. | Term Frequency–Inverse Document Frequency, Information gain | 90.89% Accuracy | - Multi-class classification is not taken into account. |
| [18] | Boosts the Ant Colony Optimization algorithm performance by using the Bayesian classification method. | None | 0.86 Avg TPR and 0.99 Avg TNR | - Comparative analysis with statistical methods, such as information gain and Chi-square, has not been explored. |
| [20] | Proposes Ant Colony Optimization for classification tasks. | None | 96.56% Accuracy | - Does propose a unique feature selection method but fails to shed light on the performance with text data. |

**Table 1.** *Cont.*

| Ref. | Summary | Feature Engineering Techniques Used | Performance Measures | Limitations |
|---|---|---|---|---|
| [10] | Proposes an embedded approach to feature selection using the Chi-square approach for filtering the features. | Term Frequency–Inverse Document Frequency, Chi-square | 89.08 Micro-F1 and 78.42 Macro-F1 | - Does not discuss the effectiveness of the approach on another textual dataset as well as its impact on the data having numerical columns. |
| [22] | Discusses a combined SVM-ACO pipeline for choosing more relevant features from large datasets. | None | 95.90% Accuracy | - Fails to cover text classification and its impact on feature selection algorithms. |
| [23] | Presents the hybridization of ACO and genetic algorithm for feature selection. | None | 99.93% Accuracy | -Multi-class classification of text data not performed, and no comparison with standard feature selection methods. |
| [24] | Proposes a hybrid strategy for feature subset selection using the classifier ensemble with ACO algorithm. | Information gain, Gain ratio | 97.33% Accuracy | -Not tested on text data and lacks comparison with standard feature selection methods. |
| [25] | Introduces FACO algorithm developed for feature selection. | FACO | 96% Accuracy | - Lacks discussion of data type used in the experiment. |
| [19] | Proposes spam classification using Naive Bayes's classifier for improved results. | Ant Colony Optimization | 84% Accuracy | - Only one classifier has been used for classification. |

## 3. Data and Methodology

The implementation of our proposed architecture was made possible using the following steps.

### 3.1. Data Acquisition

There are various datasets and corpora available worldwide in different languages or models. To fulfill our requirement of drawing a comparative analysis between textual and numerical data under the hood of Ant Colony Optimization, we looked through the various datasets that could help us. We required both textually labeled datasets and numerically labeled datasets for our research purpose. We thus chose to use Twitter as a source for extracting our text dataset for sentiment analysis with three main sentiments—positive, negative, and neutral—and the most commonly used Stroke Prediction Dataset for numerical analysis. Table 2 shows the summary of datasets describing the number of samples in the dataset, whether the dataset is balanced or not, resampling techniques used, and the number of labels and annotations used.

**Table 2.** Dataset details shown with different characteristics.

| Name of Dataset | Number of Samples | Balanced/ Unbalanced | Resampling Technique | No. of Labels | Name of Labels |
|---|---|---|---|---|---|
| Stroke Prediction (Numerical Data) | 5110 Stroke: 249 No stroke: 4861 | Unbalanced | SMOTE Oversampling | 2 | Stroke and No stroke |
| Twitter Data on Extremism (Text Data) | 93,501 Positive (2): 29,652 Negative (0): 33,880 Neutral (1): 29,969 | Balanced | None | 3 | Positive, Negative, and Neutral |

The raw data extraction phase was followed by the labeling phase, which was of utmost importance since a part of our study focuses on sentiment analysis on Twitter extracted data. The data were divided into three sentiment labels, i.e., negative, neutral, and positive, and visualized using a funnel chart, as shown in Figure 1.

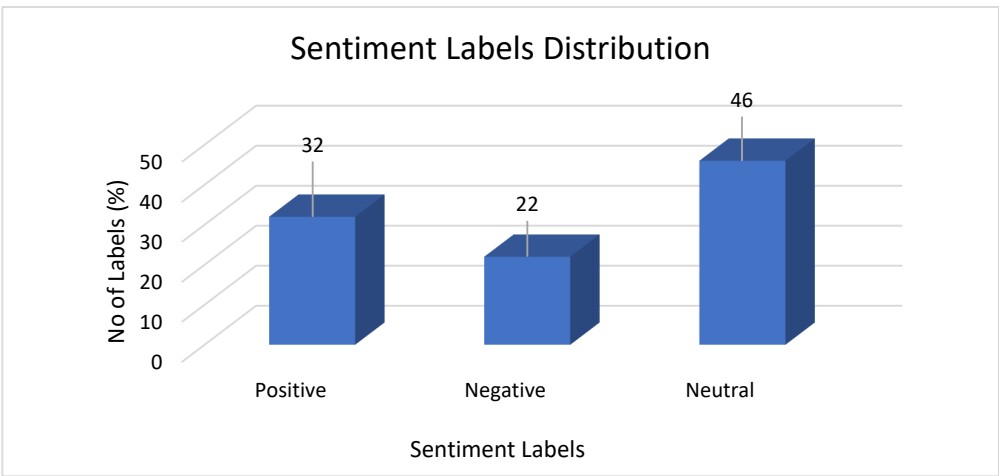

**Figure 1.** Bar chart showing the distribution of sentiment labels in the text dataset.

The dataset contains hate speech tweets prevalent during the capitol riot. To summarize the most popular words, we took the help of word cloud visualization to understand the distribution of hate words in the dataset. The tweets posted during the riot consisted of words, such as America-first, Donald trump, maga, stop Biden, etc., signifying the hatred among the users, as evident through the word cloud shown in Figure 2.

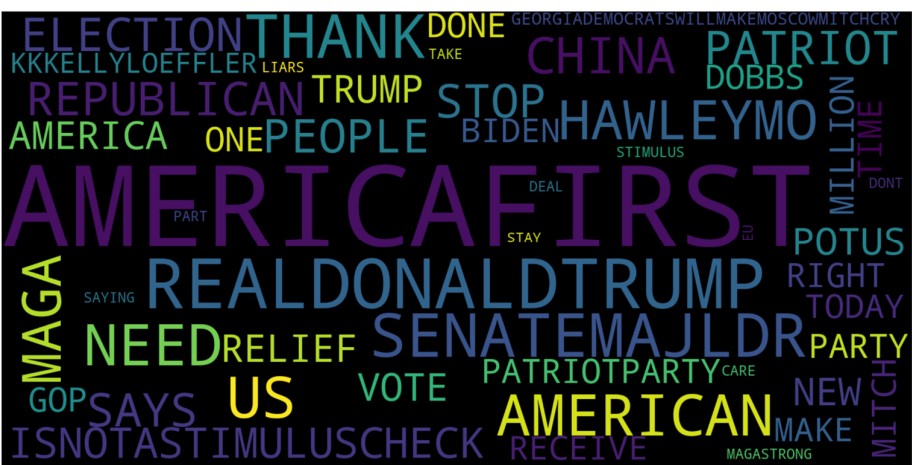

**Figure 2.** Word cloud depicts the most frequently occurring words in hate speech dataset.

On the other hand, the medical dataset curated by the World Health Organization for stroke prediction containing approximately 5110 records of potential patients and 249 occurrences of stroke was used for the numerical data analysis utilizing feature selection methods and optimization algorithms [26].

### 3.2. Data Preprocessing

After a thorough analysis of both datasets, we applied rigorous preprocessing techniques to obtain refined data. We applied techniques such as lemmatization and tokenization for the textual dataset. Similarly, we applied the oversampling technique to balance out the data for better performance for numerical data.

### 3.2.1. Stroke Dataset

For the stroke prediction dataset, preprocessing the data was essential since the raw data had a lot of null values and a high imbalance factor, which made it unsuitable for training the model. Initially, we transformed some of the columns consisting of categorical values into numerical ones through label encoding. We imputed the null values with mean

values. Moreover, the stroke prediction dataset was highly unbalanced with 5110 rows, of which 4861 records were of the 'no stroke' label, indicating a tiny sample of data with actual stroke prediction. Using such imbalanced data for training can overfit the model, ultimately giving incorrect results. Therefore, it becomes essential to appropriately deal with imbalanced data to make the findings reliable and adequate. We used a resampling strategy known as Synthetic Minority Over-Sampling Technique (SMOTE) to balance the data, as shown in Figure 3.

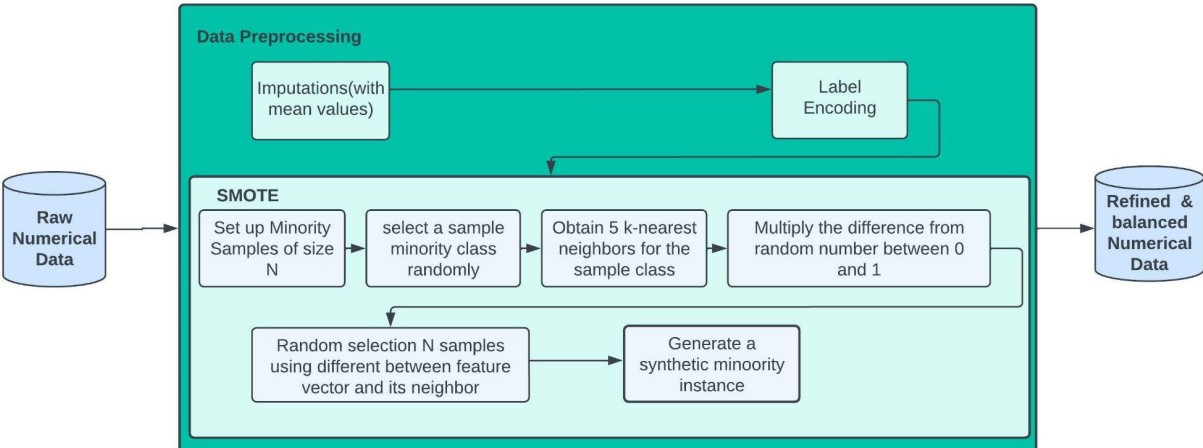

**Figure 3.** Data preprocessing steps applied to the numerical dataset.

SMOTE is a statistical approach for increasing the number of cases in a balanced manner in your dataset. The component creates new instances depending on the provided input of present minority cases.

After applying SMOTE, it does not affect the number of majority cases; instead, it affects minority cases. The algorithm produces new examples by combining the characteristics of the class label with the characteristics of its closest neighbors. This method expands the features present for each class and inflates the samples. Figure 4a,b clearly show the difference in the dataset before and after the application of SMOTE.

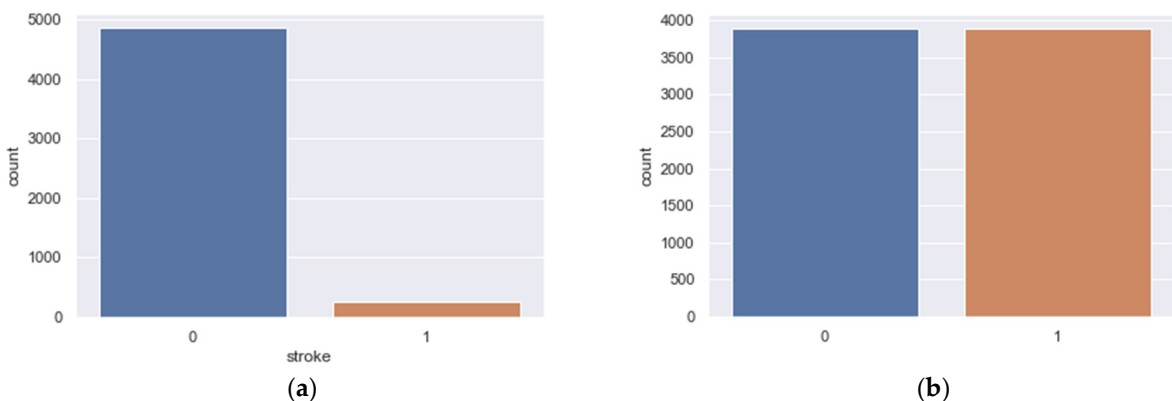

**Figure 4.** Class labels in stroke dataset (**a**) before applying SMOTE and (**b**) after applying SMOTE.

### 3.2.2. Hate Speech Dataset

For the textual dataset, data preprocessing started with converting tweets to lowercase and removing noisy data such as URLs, hashtags, HTML references, placeholders, non-letter characters (punctuation and special characters), and Twitter handles. This step was followed by stopword removal, tokenization, and lemmatization. Some of the most significant and most important preprocessing are discussed below and shown in Figure 5.

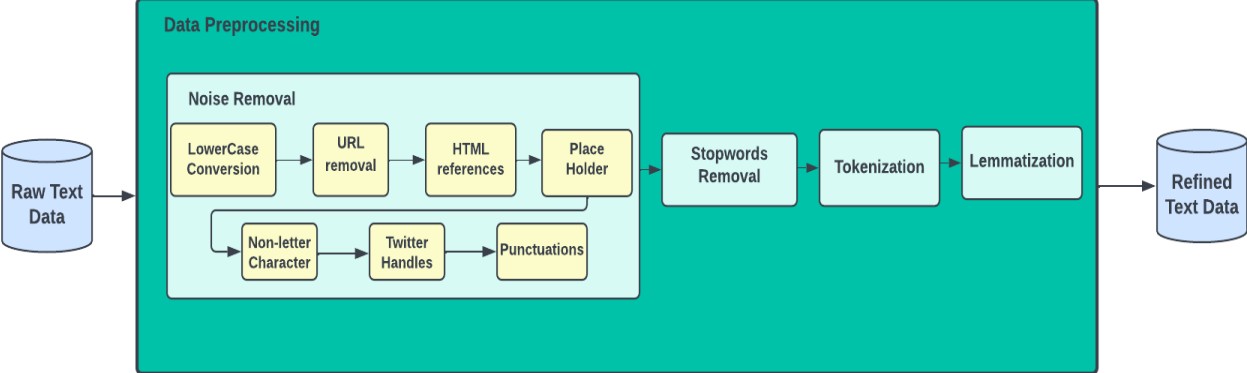

**Figure 5.** Data preprocessing steps applied to the textual dataset.

Stopword removal: These words repeatedly appear in a document and contain a small amount of information that is not usually required [27]. For example, a few words in the English language frequently appear in a text but contain little information, e.g., a, am, an, and, any, are, as, at, be, been, could, do, does, during, etc. [28]. These are known as stopwords. The term "stopword list" or "stopword corpus" refers to a collection of stopwords. Removing stopwords shrinks the vector space [29] and improves performance by increasing execution speed, calculation speed, and accuracy. Hence, removing stop words provides low information from the text and does not affect the training model [27].

Tokenization: Tokenization is dividing input text into smaller units known as tokens [30]. The computer takes the input text into one massive string of characters, in this case. After following some natural language processing steps, the tokens are given input to the system. Because these treatments are typically arranged to work on individual sentences, identifying the boundaries of sentences and tokens is frequently a subsidiary task of tokenization [31]. Therefore, tokenization helps the model to understand the context in a better manner by understanding the text semantics by its chain of words.

Lemmatization: Lemmatization refers to vocabulary and morphological analysis of words to remove articulate endings and return the base form of a word known as its lemma. This technique replaces the original word with its root word form, known as lemma [32]. A word can mean multiple things depending on its usage in a sentence. Likewise, different forms of words state the related meaning. Since they all mean the same thing, it is best to lemmatize the text, which helps improve accuracy.

*3.3. Feature Engineering Techniques*

3.3.1. Feature Extraction

Text feature extraction is critical in the classification of text as it directly impacts the accuracy of the models [33]. A classifier is unable to understand the text directly. Hence, it is necessary to convert the text into a numerical representation. We can do this by using an indexing technique to map text data into vector form. Various feature extraction techniques have been developed to achieve the vector representation of the text. Some of the most popular methods, including Term Frequency–Inverse Document Frequency and Bag of Words, have been implemented in our study to transform text data into the vector format [33].

Tf-Idf—Textual data are easier to understand in numerical values instead of text or characters by any computer language. As a result, we must vectorize all of the text to represent it better. Tf-Idf is a method to calculate the number of words in a collection of documents. *TF* and *IDF* scores are calculated individually, where *TF* calculates the frequency of words in a text, and *IDF* computes the frequency of a word in a document. The product of these two values results in stating the significance of the term in a document. Hence, a vector of the most relevant words is created to get the final set of best-performing features, as shown in Figure 6.

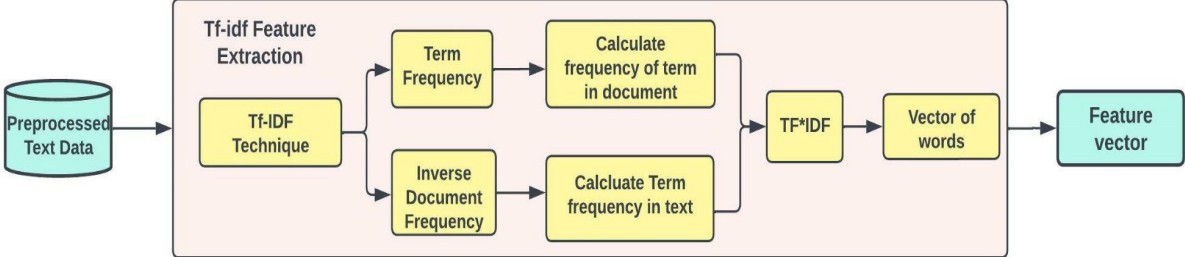

**Figure 6.** Term Frequency–Inverse Document Frequency workflow to convert text data into feature vectors.

Mathematically, we can represent it as in Equation (1):

$$TF - IDF = TF * IDF \qquad (1)$$

where *TF* is the term frequency, and *IDF* denotes the inverse frequency of the document.

Bag of Words (BoW)—The BoW model is a way of encoding text data. It is easy to learn and use and has proven effective in language modeling and document classification tasks.

The BoW represents the word count of each word that occurs in a statement. It entails two steps:

- A list of terms (vocabulary) that are well-known.
- A metric for determining the existence of vocabulary.

This approach simplifies NLP and information retrieval, where text is represented as an unordered corpus, with no respect for grammar or word order. In text categorization, weight is assigned to a word in a document based on how frequently it appears in the document and how frequently it appears in different papers. In simple words, we can state that a Bag of Words is made up of words and their weights. This study presents a new feature to the BoW model where the suggested method is evaluated to see how well this method works for text classification [34]. The working of Bag of Words is simple to understand and can be listed in three simple actions as shown in Figure 7:

1.  Tokenization of text.
2.  Building a vocabulary of words by scanning the dictionary for every token and adding them to the dictionary. If a word is not present, the frequency table is updated with a new frequency count.
3.  Lastly, text vectorization is performed to get the most frequent words in the numerical form.

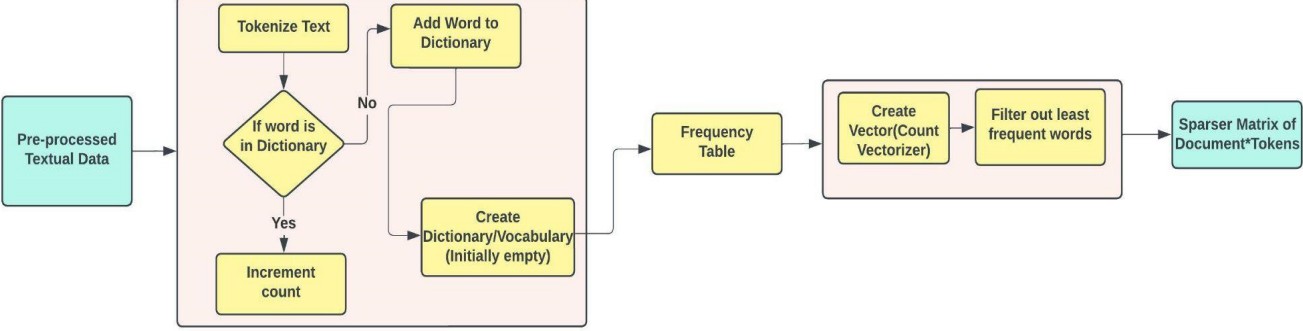

**Figure 7.** Bag of Words workflow to convert text data to feature vectors.

3.3.2. Feature Selection

To optimally train any model, we must use only essential features. If too many features are used, the model will capture irrelevant patterns leading to inaccurate training. Besides, a massive amount of data increases the training time and decreases the model's

effectiveness. Feature selection is used to solve this problem, which is two types viz. supervised and unsupervised. The supervised models can be categorized into three types: wrapper, filter, and intrinsic [35]. This study focuses on one such supervised model known as the Chi-square test, which is a type of filter method.

- Filter Method—A typical data preparation approach, the filter method combines ranking techniques with standard measures and selects variables using sorting strategies. The ranking approach removes irrelevant features before beginning the classification process [36]. Filter methods can be of two types: global and local, which depend upon whether the score is single or multi-class. A globalization policy is required in the case of local feature selection methods to combine various local values into a single global score. When using global feature selection techniques, the scores can be used to rank features directly. The features are ranked in ascending order, with the top-N features being included in the feature set, with N being an empirically determined number [37]. The feature selection method can include information gain (IG), correlation, Chi-square (CHI), Gini Index, and SVM. Chi-square is the one we are interested in for this particular study.
- Chi-square—The Chi-square test is among the most helpful statistical tests. It offers information not only on the critical difference observed, but also on the differences between categories. It helps to determine the correlation between the variables in the data. The formula employed for performing the Chi-square test could be observed as shown in Equation (2).

$$X_c^2 = \sum \frac{(O_i - E_i)^2}{E_i} \tag{2}$$

where $c$ is the degree of freedom, and $O$ and $E$ are the observed and the expected value, respectively [38].

The number of variables responsible for the change in a statistical calculation is represented by degrees of freedom. The Chi-square test can determine the degree of freedom to verify its statistical validity. These tests are widely used for comparing the observed data with predicted ones in case a particular hypothesis is correct. Chi-square will have a value of 0 when the feature is independent of the class. The main steps comprising the whole Chi-square test can be summarized as shown in Figure 8.

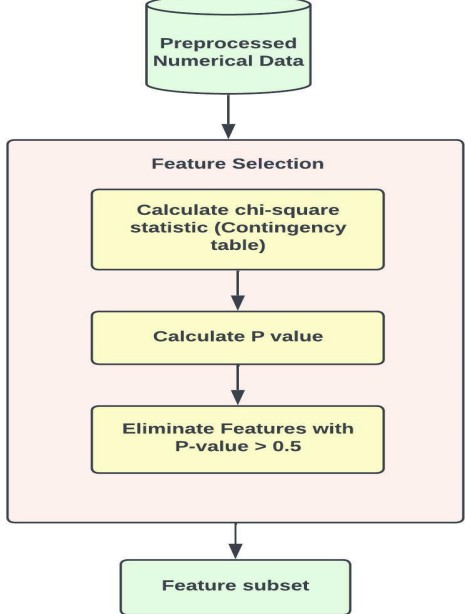

**Figure 8.** Chi-square filter method workflow to select feature subset from numerical dataset.

This test can be used for two scenarios:

1. Check data distribution according to a well-known theoretical probability distribution such as the Normal or Poisson distribution.
2. The Chi-squared test can evaluate the quality of fit of your trained model on the train, validation, and test dataset.

## 4. Machine Learning Models

### 4.1. Logistic Regression

The logistic regression model is a commonly used statistical model for determining whether independent variables affect a binary or multi-class dependent discrete variable. This particular machine learning model is often utilized for predictive analytics and modeling [39]. Its importance as a statistical and data mining technique encourages statisticians and researchers to employ this model [40]. Moreover, this type of analysis allows one to predict the likelihood of an event [41,42] and explain the relationship between independent variables and a binary outcome [43]. The logistic regression model is based on a sigmoid function wherein, for a given input, the output ranges between 0 and 1. Lastly, one of the most critical parameters of this particular model is the random state used to control the randomized nature of splitting data into train and test and to set the seed for a random generator to ensure that the generated results can be reproduced.

### 4.2. K-Nearest Neighbor

The KNN algorithm is a simple, lazy-learning, non-parametric algorithm for classification problems [44]. The algorithm performs a similarity measure to distinguish new data points with how their neighbors are classified [45]. Beginning with the initialization of parameter k, i.e., the number of neighbors, it sets the neighbor points in the original data closer to the new data point [46]. The new data point is categorized according to the majority vote of the number of neighbor points in each category [47]. This algorithm calculates the distance between the new data point to each neighbor point and classifies the instances to select the nearest neighbor.

### 4.3. Random Forest

RF is a supervised learning technique that is adaptable, simple to use, and does not generate hyper-parameters, thereby performing well in classification problems. In Random Forest, the number of trees generated is restricted for all the data to be categorized. The breaker properties affect the total number of trees for each data point, significantly impacting the accuracy. The number of trees can be increased to improve the model's accuracy; however, we start with a small number of trees. The model will be less accurate if the total breaker characteristics are equal to the total attributes accessible [36]. Random Forest (RF) models are made up of various decision trees. The feature space is divided into a certain number of regions using a decision tree with the same number of leaves [48]. For classification, the outcome of each decision tree is recognized based on majority voting. The prediction power of the model depends upon specific hyperparameters, such as n_estimators, which set the number of trees built by the algorithm before the averaging process. However, the model uses no specific formula to predict the outcome.

### 4.4. Stochastic Gradient Descent

Robbins and Monro proposed the Stochastic Gradient Descent (abbreviated as SGD) model in their paper "A Stochastic Approximation Method" [49]. Gradient descent is a widely used optimization technique that can be used in practically any learning algorithm. A function's gradient is its slope, which tells how much one variable changes in response to changes in another. Gradient descent is a convex function with a mathematical definition whose output is the partial derivative of a set of input values. SGD is a variant of gradient descent in which only a few samples are chosen at random for each iteration rather than the entire dataset [49,50].

## 5. Ant Colony Optimization Algorithm

### 5.1. Optimization Algorithm

Optimization is used in almost every field, from engineering to economics. Because resources and time are permanently restricted, making the use of what is available is most critical. Under various complex constraints, most optimizations in the real world are highly non-linear and multimodal [51]. Different goals are frequently at odds. Even for a single goal, there are situations when optimal solutions do not exist. Hence, many engineering optimization problems are notoriously difficult to solve today, and numerous applications must deal with them. Finding an ideal or even sub-optimal solution is not an easy undertaking. The search space in these situations grows exponentially in proportion to the magnitude of the problem. Therefore, typical optimization methods do not offer them an appropriate answer. As a result, various computational intelligence paradigms, which are called metaheuristic algorithms, have been developed to handle complex optimization problems over the last few decades [51,52]. The authors of [53,54] studied the optimization techniques in deep learning. Our implementation aims to draw a comparative analysis between textual and numerical data performance with a primary focus on one such metaheuristic algorithm, known as Ant Colony Optimization.

### 5.2. Ant Colony Optimization

ACO is one of the metaheuristic approaches for solving complex combinations of optimization problems [55]. It is purely inspired by the hunting technique of ant colonies, which communicate through various means [24,56]. It comes under the broad category of nature-inspired intelligence technique; swarm intelligence is based on the collective behavior of swarms having self-organized nature [57]. Even though ACO has attracted many researchers due to its ability to tackle various complicated issues and provide optimum solutions, it has been chiefly productive in numerical data, unlike textual data, where it faces many challenges.

### 5.3. Architecture

Ant Colony Optimization follows a heuristic strategy where the flow of the algorithm is further subdivided into several steps. First, we have significant parameters affecting the efficiency and accuracy of complex problems. These include several ants and epochs that need to be initialized, followed by the generation of a global random variable required to calculate the fitness score. This fitness score determines how well our solution fits the problem in consideration [58]. Our model had additional parameters, such as alpha and beta, providing relative importance to learning and heuristics. Using ACO, the optimization problem is reduced to finding the optimum path on a weighted graph. By traveling over the network, the artificial ants incrementally create solutions. The solution creation process is stochastic and is influenced by the generation and revision of the pheromone model or matrix, a parameter set connected to graph components. The ants alter the values of this parameter set at runtime [59]. A transition function is applied to generate a new path, and the whole process repeats itself for n number of iterations [60]. These steps finally generate a feature set with the best fitness score and are trained with different ML models for better accuracy, as shown in Figure 9.

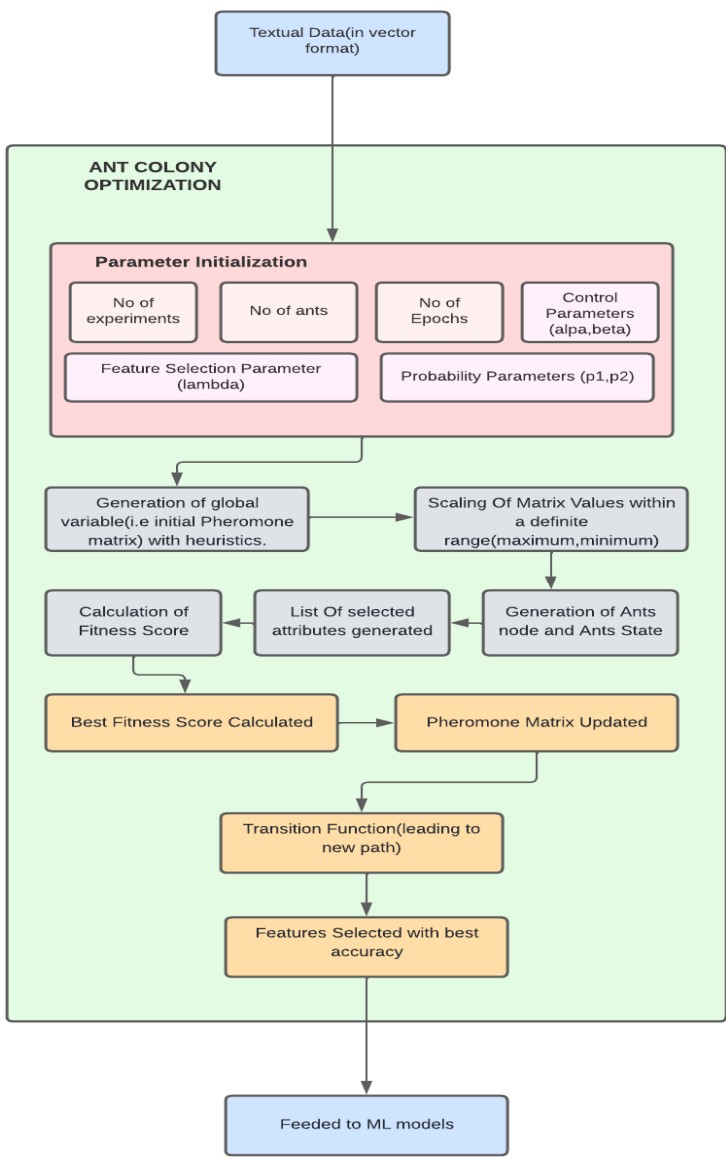

**Figure 9.** Ant Colony Optimization architecture for feature selection.

## 6. Proposed Architecture

Our proposed architecture consists of a novel framework comprising feature extraction and an optimization algorithm for feature selection techniques applied independently to textual and numerical data, as shown in Figure 10. This proposed system aims to determine the effect of Ant Colony Optimization (a metaheuristic approach) by selecting features to show a comparative analysis of both numerical and textual datasets and to study improvement in the performance of machine learning models.

The comparison was conducted on two separate datasets, one being textual and the other being numerical. The textual data were extracted from Twitter in their raw form, requiring conversion into the vector format to be optimized by any optimization algorithm. Initially, the whole dataset was refined and cleaned using different preprocessing techniques such as lemmatization and tokenization. This process was followed by feature extraction approaches, such as TF-IDF and Bag of Words, to isolate the most vital features. TF-IDF was utilized as the weighting function to assign weights to words in the text depending on their frequency calculated by vectorizing the text [61]. On the other hand, the textual data were converted into a vector format using the Bag of Words techniques to depict the occurrence of each word in a sentence [34].

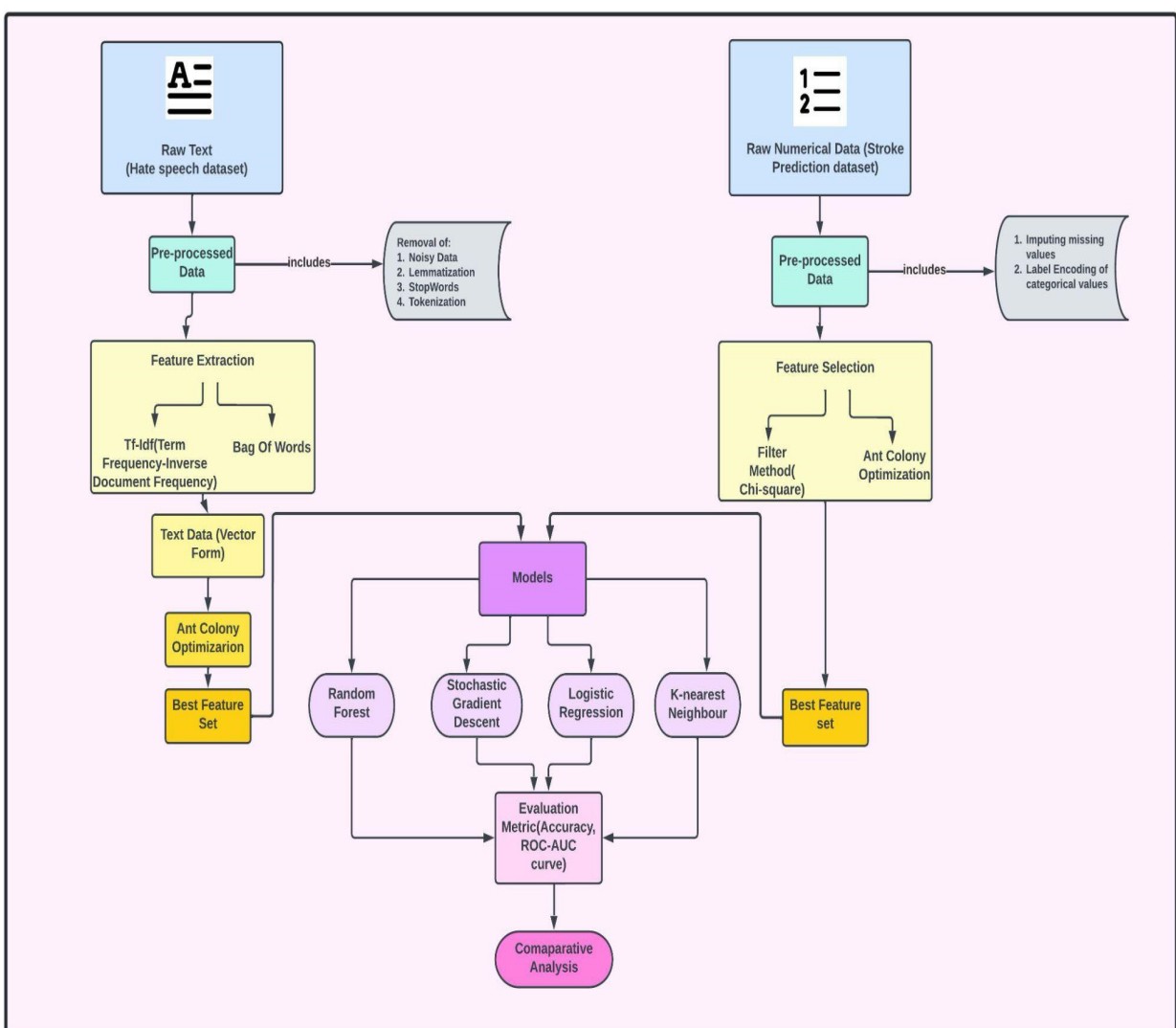

**Figure 10.** Proposed architecture with Ant Colony Optimization (ACO) applied on different machine learning algorithms.

The feature space's vast dimensionality is a fundamental challenge in text categorization. This challenge makes feature extraction a critical phase in classification tasks. There are numerous approaches to dealing with text feature extraction at the moment. This research proposes an approach based on ACO to increase the performance of text categorization and compare its efficiency with numerical data to understand the challenges faced in text classification [62]. We fed the textual data obtained from the feature extraction techniques separately to the Ant Colony Optimization algorithm. The process produces a set of features with the best fitness score and accuracy, which were then trained using four unique machine learning models, namely, LR, KNN, RF, and SGD.

The 'stroke prediction' dataset was used to study the performance of ACO on numerical data. The data were preprocessed by removing the noise, outliers, and missing values to make them suitable for model training. We identified the columns with null or missing values and performed imputations using mean values. Since the dataset had categorical values, we handled them through label encoding. As a result, after label encoding, the string values in the entire dataset were converted into respective numerical representations [63]. Using the feature selection technique, we used the preprocessed data to find the set of best-performing features. Firstly, the Chi-square approach was applied for feature selection. The Chi-Square test determines if the independent variables support either the

null or alternative hypothesis. In the null hypothesis, the predictors and the outcome variable are independent, whereas they are dependent on the alternative hypothesis. The feature or variable is chosen if the alternative hypothesis is true; otherwise, the feature is ignored [62]. Similarly, a set of features was selected using the ACO algorithm. Then, we trained the models (RF, KNN, LR, and SGD) using these features to analyze the difference in the performance of the traditional and optimized feature selection methods.

Our proposed architecture used accuracy and ROC-AUC scores and curves as an evaluation metric to compare the models based on their performance. Lastly, we performed a comparative analysis on both datasets to figure out the differences in the result of numerical and textual feature optimization.

## 7. Experimental Setup

### 7.1. Dataset

The text dataset was built from scratch using hashtags related to the US Capital riot on Twitter. The collection was performed using the Twitter-scraper library [64]. The data were collected from 25 December 2020 to 10 January 2021, with specifically English tweets extracted. Keywords and search filters were applied to clean off irrelevant tweets.

The dataset mainly had five columns:

- Keyword: the keyword used for the extraction of the tweets;
- Datetime: the date on which the tweet was posted;
- Tweet-id: unique id for each tweet posted;
- Text: it contained the actual tweets posted by the user;
- Username: it contained the name of the user who posted the tweet.

### 7.2. Parameter Settings

Most metaheuristic algorithms used for feature selection have one or more hyperparameters. We must initialize these parameters before conducting experiments. The authors selected parameters of ACO based on the experiments referring to [65]. This section discusses the parameter settings of the ACO algorithm for the experiments conducted on different feature extraction methods. To begin with, we used a textual dataset where we implemented the TF-IDF and Bag of Words feature extraction method and assigned unique parameter values in ACO for each of them. As shown in Table 3, for the experimental setup conducted with TF-IDF, we took the number of ants as 40. The ACO algorithm works with N number of ants that selects features and creates feature subsets from the original feature set. Furthermore, the variables alpha and beta are of relative importance to learning, i.e., the pheromone and heuristic set as 1 and 0.2. We performed this experiment in one iteration for two epochs.

**Table 3.** Parameter settings of Ant Colony Optimization algorithm.

| Dataset | Feature Extraction Technique | Optimization Algorithm | Population | Iteration | Epoch | Alpha | Beta |
|---|---|---|---|---|---|---|---|
| Hate Speech dataset | TF-IDF | Ant Colony Optimization | 40 | 1 | 2 | 1 | 0.2 |
| | Bag of Words | Ant Colony Optimization | 80 | 1 | 2 | 1 | 0.02 |
| Stroke dataset | SMOTE | Ant Colony Optimization | 20 | 1 | 5 | 1 | 0.2 |

Meanwhile, for the Bag of Words experiment, we set the number of ants as 80. Similarly, the alpha parameter was set as 1, while the beta parameter was set as 0.02. In this experiment, the number of iterations was one, and the number of epochs was two. On the numerical dataset, we performed SMOTE along with ACO, where we took the number of ants as 20, and the alpha and beta parameters remained 1 and 0.2, respectively, as per the TF-IDF experiment. However, the iteration count is kept as one, and the count of epochs equaled five. A Random Forest classifier is incorporated within the Ant Colony

Optimization algorithm, which helps to evaluate the fitness of each feature subset created by ants. In return, it gives the best fitness score, best accuracy, and selected features set. To assess the robustness of each set of selected features by each ant, we employed a Random Forest classifier that returns the best fitness score, best accuracy, and the number of features selected.

*7.3. Evaluation Metrics*

A model's performance can be estimated using evaluation metrics since they can discern between different model results. Basic evaluation metrics include precision, recall, f-measures, and accuracy, commonly employed to forecast job efficiency. These measures are used to examine the performance efficiency in classification models. For this study, we compared our models based on two metrics—accuracy and the AUC-ROC curve. These metrics are discussed as follows.

Accuracy—Accuracy is one of the most straightforward measures to calculate and comprehend the result of machine learning models. It can be defined as the ratio of the sum of the true positive and true negative of the actual data. The true positive signifies a successfully predicted event, whereas the false positive indicates the wrongly predicted events [66]. Similarly, true negative indicates the successfully predicted no-event values. False negative denotes the no-event values predicted mistakenly [66]. We can write the formula as Equation (3):

$$Accuracy(a) = \frac{TP + TN}{TP + FP + TN + FN} \tag{3}$$

In the formula above, *TP*, *TN*, *FP*, and *FN* represent true positive, true negative, false positive, and false negative, respectively.

AUC-ROC Curve—The Receiver Operator Characteristic, popularly called ROC, is a true positive rate (*TPR*) plot versus false positive rate (*FPR*) at all possible threshold values [67]. Another metric called area under the ROC curve (AUC) is used to measure the points in the curve. The area under the curve suggests the capacity of a classifier to differentiate between classes. Since our study involves binary and multi-class classification, this metric proves helpful in determining the best-performing classifier. The formula used to calculate the *TPR* and *FPR* is as written Equations (4) and (5):

$$TPR = \frac{TP}{TP + FN} \tag{4}$$

$$FPR = \frac{FP}{FP + TN} \tag{5}$$

## 8. Results and Discussion

This section discusses the results obtained from the different methodologies and pipelines implemented and compares them based on a solid foundation. To start with an overview, we analyzed two datasets, hate speech and stroke prediction. These datasets were initially preprocessed using suitable techniques according to the dataset's data type, followed by different feature engineering techniques, such as TF-IDF, BoW, and Chi-square test, to manipulate the data efficiently. Optimizing the features could be seen as the next step, where we adopted the metaheuristic approach and implemented ACO to present the importance of optimization algorithms in terms of efficiency. Lastly, four ML models were chosen to train the dataset: Random Forest, Stochastic Gradient Descent, Logistic Regression, and K-Nearest Neighbor, and the proposed models were evaluated based on accuracy and ROC curves to present and review a comparative study.

The summary results for our experiments are shown in Table 3 for both datasets. We can observe that the Ant Colony Optimization algorithm has a significant advantage in model training and testing when applied to both textual and numerical datasets for all the machine learning models implemented. We can see that ACO increases the efficiency of

models by a maximum of 10.07%, with Random Forest giving the highest accuracy of 80% for the TF-IDF feature extraction technique. Simultaneously, models such as SGD, LR, and KNN achieve an accuracy of 60%, 80%, and 60%, with the percentage increase in accuracy being in the range of 0.17%, 0.67%, and 3.62%, respectively. Likewise, Bag of Words combined with ACO gives a maximum accuracy of 80% with the Random Forest model and 17%, 15%, and 7% efficiency increments for SGD, LR, and KNN models, respectively.

A similar comparison was performed for the numerical dataset, where we applied the filter method of the feature selection technique and received 88.6% as the highest accuracy for the Random Forest classifier. The accuracy score increased by 1.15% when we optimized the feature selection using ACO, proving that using optimization algorithms does help in improving the performance of the models. The other three models, LR, KNN, and SDG, similarly outperformed when trained with feature selection using ACO with an accuracy of 76.8%, 84.6%, and 76.25%, respectively. The difference in the performance of the models with and without ACO can be observed in Table 4. The table clearly shows a marginal increase of 0.2%, 1.6%, and 0.35% in the respective models' efficiency.

**Table 4.** Dataset evaluation with and without ACO algorithm.

| Dataset | Feature Engineering Technique | Optimization Algorithm | Models | Accuracy |
|---|---|---|---|---|
| Hate Speech Dataset (Textual) | Tf-idf | ACO | **Random Forest** | **80%** |
| | | | **LR** | **80%** |
| | | | SGD | 60% |
| | | | KNN | 60% |
| | BoW | ACO | **Random Forest** | **80%** |
| | | | **SGD** | **80%** |
| | | | LR | 80% |
| | | | KNN | 60% |
| | Tf-idf | None | **LR** | **79.33%** |
| | | | Random Forest | 69.93% |
| | | | SGD | 59.83% |
| | | | KNN | 56.38% |
| | BoW | None | **Random Forest** | **66%** |
| | | | LR | 65% |
| | | | SGD | 63% |
| | | | KNN | 53% |
| Stroke Prediction Dataset (Numerical) | SMOTE | ACO | **Random Forest** | **98.43%** |
| | | | SGD | 92.76% |
| | | | LR | 97.06% |
| | | | KNN | 97.16% |
| | SMOTE + Filter Method (chi-square) | None | **Random Forest** | **88.6%** |
| | | | KNN | 83% |
| | | | LR | 76.6% |
| | | | SGD | 75.9% |

Additionally, when analyzed in-depth, the observed experimental results clearly show the difference in the performance of models for both datasets independently. When trained on the numerical dataset and optimized through ACO, we can observe that the models

perform better by approximately 20.75% than the textual dataset, indicating the challenges and limitations faced while working with text features.

Visualization is vital for sorting out various aspects and anticipating marginal contributions that may be used to assess the feature importance across different models in terms of evaluation metrics such as accuracy and the ROC-AUC curve. Since our study focuses on comparative analysis, we visualized the model performances. Figure 11a shows the accuracy achieved by the four models-RF, LR, SGD, and KNN, when trained with textual and numerical data and enhanced using ACO. The machine learning models outperform in terms of efficiency when fed with numerical data. Figure 11b further shows the performance of the same models but without any optimization algorithm. Both the figures, when compared together, establish the contribution and advantage of Ant Colony Optimization in our proposed analysis.

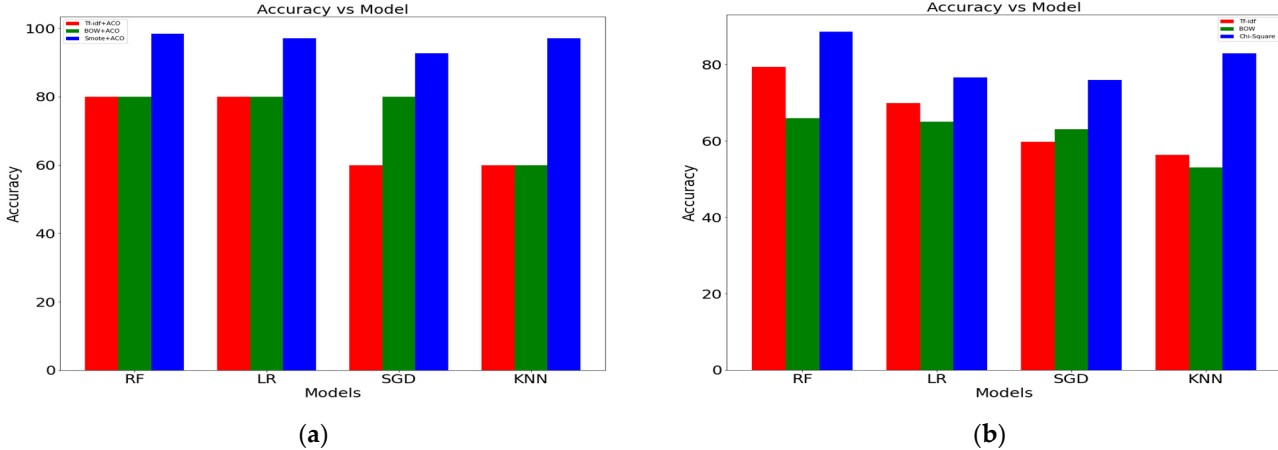

(**a**)   (**b**)

**Figure 11.** Accuracies of ML models on textual and numerical datasets (**a**) using ACO and (**b**) without using ACO.

AUC-ROC statistics aid in determining a model's capacity to discriminate across classes [50]. Hence, we decided to offer the experimental results in the most generally used evaluation measure, clarifying why we chose such a pipeline. The ROC-AUC curve was plotted for the best models integrated with ACO on textual and numerical datasets. The ROC curves were plotted for Random Forest classifier on ACO, using TF-IDF and BoW as the effective feature extraction techniques. The curve proves that the proposed model is sufficient to correctly predict the target labels since the area under the curve is significantly large, as shown in Figure 12a,b. Moreover, we can observe that the area underclass positive is highest for ACO on textual data with TF-IDF and almost equal for neutral and negative, which is directly proportional to the prediction level of the model for each target label. The ROC-AUC scores obtained for TF-IDF and BoW are 0.88 and 0.77, respectively, which indicate that the models are moderate classifiers.

On the other hand, the classifier for the numerical dataset gave a ROC-AUC score of 0.98, which is approximately 10.1% greater than the textual dataset. The plotted ROC curve shown in Figure 13 further proves the outstanding performance in terms of area under the curve of random forest classifier over any other algorithm such as KNN, LR, or SGD. It is noticeable from Table 5 that benchmark accuracies were achieved for both datasets. For textual data with TF-IDF as the feature extraction technique, Random Forest and Logistic Regression outperformed all other models with an accuracy of 80% and a ROC-AUC score of 0.88. In contrast, when fed to the models, text features extracted using Bag of Words performed best with Random Forest and Stochastic Gradient Descent with the same accuracy and a ROC-AUC score of 0.77. The highest figures obtained for numerical data belonged to the Random Forest classifier, where we achieved an accuracy of 98.43%. When it comes to the ROC-AUC score, RF was again the leading algorithm with a value of 0.98. Hence, a clear difference in performance and efficiency can be observed wherein

machine learning models underperformed by approximately 18.43% on text data using ACO compared to when trained on numerical data.

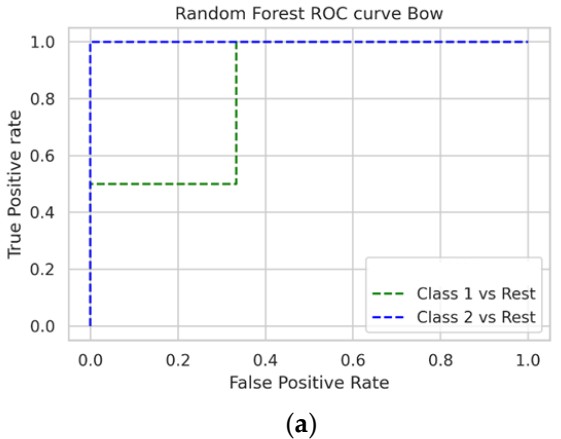

(**a**)

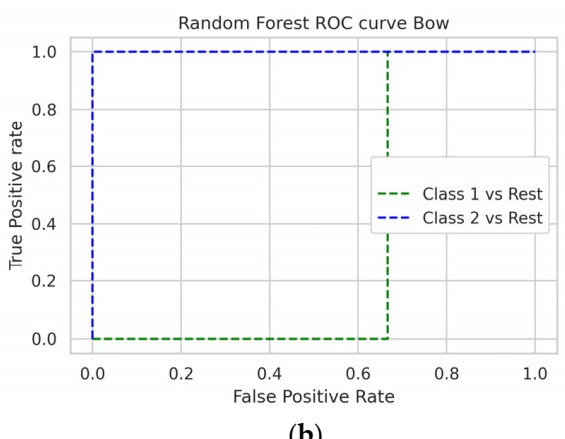

(**b**)

**Figure 12.** ROC curve for ACO on textual data (**a**) TF-IDF and (**b**) BoW.

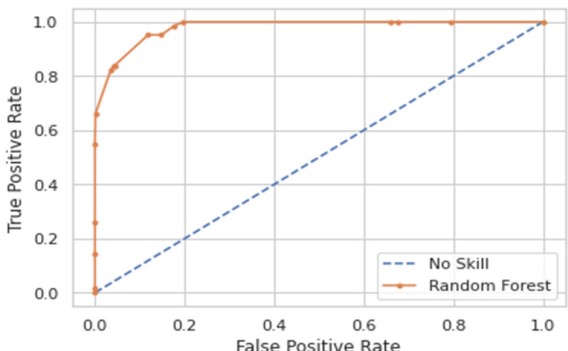

**Figure 13.** ROC curve for a numerical dataset with ACO.

**Table 5.** Performance comparison of best models for textual and numerical dataset.

| Model | Methodology | Dataset Type | Accuracy | ROC-AUC Score |
|---|---|---|---|---|
| Random Forest, Logistic Regression | TF-IDF + ACO | Text | 80% | 0.88 |
| Random Forest, Stochastic Gradient Descent | Bag of Words + ACO | Text | 80% | 0.77 |
| Random Forest | SMOTE + ACO | Numerical | 98.43% | 0.98 |

Challenges and Limitations—Significant associated techniques and developments have been made in the text classification field. However, there is still room for improvement in various sectors, ranging from the accuracy of systems to their inclusivity. Textual data vary widely in size. Even though a significant amount of emphasis has been put on text classification, a lack of standard size in textual data is still a huge issue. Machine learning algorithms perform efficiently with fixed-length inputs and outputs, posing another problem for textual datasets with variable input and output sizes.

Moreover, data in text format, being unstructured, require a great deal of refinement and preprocessing, increasing the computational cost [68]. Since machine learning algorithms cannot work with raw text, it becomes necessary to convert the data into numerical form [68]. During this transformation of textual features into vectors, the complexity of data increases, making it difficult to interpret the original text's meaning, ultimately affecting the performance of the models [69]. Textual data present additional obstacles, such as improper spelling and sentence structure, making it difficult to extract the relevant information and interpret it [70]. Moreover, small, labeled samples and high dimensionality

are becoming common text classification problems [71,72]. Since labeling texts necessitates human engagement, the number of un-labeled texts grows faster than the number of labeled texts [70].

Further, real-time text data bring real-time obstacles to the table in the form of computational power and time. The larger the size of the text data, the more significant the computational complexity and the longer the computational time, which brings the performance of the models down to one fact: the capacity of the system to handle real-time text data. Attempts to make text classification dynamic are still going on; however, percent success has yet to be achieved.

### 9. Conclusions and Future Scope

Feature selection and feature extraction hold a strong foundation in preprocessing text data that directly impacts the result of any experiment. In this study, we demonstrated the use of ACO for optimized feature selection on both numerical and textual datasets. The algorithm proves its dominance over other approaches by showing a significant increase in the performance of four different machine learning models. The results after the application of ACO were tested for both the datasets (numerical and textual), where a notable increase of almost 10.07% was noticed for the ML models. The numerical data outperformed the textual data with a difference of almost 20.75% in terms of accuracy, proving that numerical features offer notable advantages when it comes to the model's performance. The experiment was conducted against multiple feature selection techniques (TF-IDF, Bag of Words, and filter method) for comparison purposes as well as to understand the challenges of text data that impact the performance of models. The experiment clearly highlights the importance of ACO with an equal focus on the processing challenges faced by the text data.

The results can be improved by including more data as this study performs the experiment on the portion of datasets due to the limited computational resources. Moreover, our work can be extended by analyzing similar other techniques like the wrapper and hybrid methods. The performance of deep learning models can also be observed to conduct an advanced study of the performed experiments. The limitations of text data can be outdone, and other optimization algorithms can be studied to understand their impact on the performance of the models.

**Author Contributions:** Conceptualization S.G., D.D. and S.P.; methodology M.Y., S.B. and A.R.; software M.Y., S.B. and A.R.; validation S.G. and S.P.; formal analysis D.D.; investigation, resources S.G. and S.P.; data curation S.G. and D.D.; writing—original draft preparation M.Y., S.B. and A.R.; writing—review and editing S.G. and D.D.; visualization S.P.; supervision K.K.; project administration S.G. and K.K.; funding acquisition D.D. All authors have read and agreed to the published version of the manuscript.

**Funding:** This research received no external funding.

**Data Availability Statement:** Not applicable.

**Conflicts of Interest:** The authors declare no conflict of interest.

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
