# Peer review of "Textual Feature Extraction Using Ant Colony Optimization for Hate Speech Classification"

_2504-2289, doi:10.3390/bdcc7010045_

Round 1
Reviewer 1 Report
I suggest the following changes to the paper:
1. What does "performance" mean in abstract? "An improvement in performance of 10.07% is observed for all the models on the application of ACO.". Authors should better explain this sentence or change it.
2. In the first part of the introduction, from line 38 to 56, many of your statements need citations. Same from line 82 to 92.
3. In Sections 3, 4 and 5 there is a lot of well-known theory, the authors should better highlight the novelties of their method and how they were used.
4. Please, consider these works in your paper:
doi.org/10.1007/s12559-022-10084-6
doi.org/10.1016/j.eswa.2022.119391
5. The authors present the dataset in Section 6. Is their architecture suitable only for the presented dataset? Or it is just used for their experiments? Please, clarify this part.
6. Tables are extremely large and difficult to read.
7. Please re-read and check for typos. For example:
* "this paper intents to bridge", Introduction
Author Response
Dear Reviewer,
Thank you for the suggestions,
We have modified the manuscripts according to your comments.
Kindly consider.
- What does "performance" mean in the abstract? "An improvement in the performance of 10.07% is observed for all the models on the application of ACO.". Authors should better explain this sentence or change it.
Author Response 1: Thank you for your suggestion. The authors have added lines in the abstract indicating the accuracy, as performance measures presented in the paper.
“A maximum improvement in accuracy of 10.07% is observed for Random Forest with TF-IDF feature extraction technique on the application of ACO.”
- In the first part of the introduction, from line 38 to 56, many of your statements need citations. Same from line 82 to 92.
Author Response 2: Thank you for your suggestion. The authors have added citations for statements in the paper. The authors have added citations from [61] to [76].
[61] Linardatos, P.; Papastefanopoulos, V.; Kotsiantis, S. Explainable AI: A Review of Machine Learning Interpretability Methods. Entropy 2021, 23, 18. https://doi.org/10.3390/e23010018
[62] Ahmad, S.R., Bakar, A.A., & Yaakub, M.R. (2019). Ant colony optimization for text feature selection in sentiment analysis. Intell. Data Anal., 23, 133-158.
[63] Ting, T. O., Saraç, Esra, Özel, Selma AyÅŸe, 2014, "An Ant Colony Optimization Based Feature Selection for Web Page Classification", - https://doi.org/10.1155/2014/649260, The Scientific World Journal, Hindawi Publishing Corporation
[64] Fan C, Chen M, Wang X, Wang J and Huang B (2021) A Review on Data Preprocessing Techniques Toward Efficient and Reliable Knowledge Discovery From Building Operational Data. Front. Energy Res. 9:652801. doi: 10.3389/fenrg.2021.652801
[65] Kumar, Sachan Rohit and Kushwaha Dharmender Singh. “Nature-Inspired Optimization Algorithms: Research Direction and Survey.” (2021).
[66] Rodrigues, D., Yang, XS., de Souza, A.N., Papa, J.P. (2015). Binary Flower Pollination Algorithm and Its Application to Feature Selection. In: Yang, XS. (eds) Recent Advances in Swarm Intelligence and Evolutionary Computation. Studies in Computational Intelligence, vol 585. Springer, Cham. https://doi.org/10.1007/978-3-319-13826-8_5
[67] Hema Banati and Monika Bajaj, Fire Fly Based Feature Selection Approach, IJCSI International Journal of Computer Science Issues, Vol. 8, Issue 4, No 2, July 2011
[68] Amelio, A., Bonifazi, G., Corradini, E. et al. A Multilayer Network-Based Approach to Represent, Explore and Handle Convolutional Neural Networks. Cogn Comput (2022). https://doi.org/10.1007/s12559-022-10084-6
[69] Alessia Amelio, Gianluca Bonifazi, Francesco Cauteruccio, Enrico Corradini, Michele Marchetti, Domenico Ursino, Luca Virgili, Representation and compression of Residual Neural Networks through a multilayer network based approach, Expert Systems with Applications, Volume 215, 2023, 119391, ISSN 0957-4174, https://doi.org/10.1016/j.eswa.2022.119391.
[70] Lundberg, S. M., Nair, B., Vavilala, M. S., Horibe, M., Eisses, M. J., Adams, T., ... & Lee, S. I. (2018). Explainable machine-learning predictions for the prevention of hypoxaemia during surgery. Nature biomedical engineering, 2(10), 749-760.
[71] Najafabadi, M. M., Villanustre, F., Khoshgoftaar, T. M., Seliya, N., Wald, R., & Muharemagic, E. (2015). Deep learning applications and challenges in big data analytics. Journal of big data, 2(1), 1-21.
[72] Gao, S., & Ng, Y. K. (2022). Generating extractive sentiment summaries for natural language user queries on products. ACM SIGAPP Applied Computing Review, 22(2), 5-20.
[73] Gao, S., & Ng, Y. K. (2022). Generating extractive sentiment summaries for natural language user queries on products. ACM SIGAPP Applied Computing Review, 22(2), 5-20.
[74] Wei, Xianmin. (2014). Parameters Analysis for Basic Ant Colony Optimization Algorithm in TSP. International Journal of u- and e-Service, Science and Technology. 7. 159-170. 10.14257/ijunesst.2014.7.4.16.
[75] Shima Kashef, Hossein Nezamabadi-pour,An advanced ACO algorithm for feature subset selection,Neurocomputing,Volume 147,2015,Pages 271-279,ISSN 0925-2312,https://doi.org/10.1016/j.neucom.2014.06.067.
[76] H. S. Alghamdi, H. L. Tang and S. Alshomrani, "Hybrid ACO and TOFA feature selection approach for text classification," 2012 IEEE Congress on Evolutionary Computation, Brisbane, QLD, Australia, 2012, pp. 1-6, doi: 10.1109/CEC.2012.6252960.
- In Sections 3, 4 and 5, there is a lot of well-known theory; the authors should better highlight the novelties of their method and how they were used.
Author Response 4: Thank you for your suggestion. The authors have added a table in the paper highlighting the novelties of their methods.
Table 1. Relevant research papers were studied in the literature review with different perspectives.
- Please, consider these works in your paper:
doi.org/10.1007/s12559-022-10084-6
doi.org/10.1016/j.eswa.2022.119391
Author Response 4: Thank you for sharing the related article. We have included articles in the manuscript.
[68] Amelio, A., Bonifazi, G., Corradini, E. et al. A Multilayer Network-Based Approach to Represent, Explore and Handle Convolutional Neural Networks. Cogn Comput (2022). https://doi.org/10.1007/s12559-022-10084-6
[69] Alessia Amelio, Gianluca Bonifazi, Francesco Cauteruccio, Enrico Corradini, Michele Marchetti, Domenico Ursino, Luca Virgili, Representation and compression of Residual Neural Networks through a multilayer network based approach, Expert Systems with Applications, Volume 215, 2023, 119391, ISSN 0957-4174, https://doi.org/10.1016/j.eswa.2022.119391.
- The authors present the dataset in Section 6. Is their architecture suitable only for the presented dataset? Or is it just used for their experiments? Please, clarify this part.
Author Response 5:
The architecture presented in Section 6 is not generic architecture, it is customized for the dataset used in the article. But we would like emphasis that the proposed architecture is unique due to the incorporation of Ant Colony Optimization (ACO) in the feature selection phase along with additional filter wrapping methods.
The following sentences are included in the paper on page number 14.
“Our proposed architecture consists of a novel framework comprising feature extraction and an optimization algorithm for feature selection techniques applied independently to textual and numerical data, as shown in figure 10. This proposed system aims to determine the effect of Ant Colony Optimization (a meta-heuristic approach) by selecting features to show a comparative analysis of both numerical and textual datasets and to study improvement in the performance of machine learning models.”
- Tables are extremely large and difficult to read.
Author response: Thank you for the suggestion. The authors have made changes in Table 1 and presented it in a concise manner. Please refer to table 1 on page number 4. Table 1 highlighted with yellow color.
Table 1: Relevant research papers studied in the literature review with different perspectives.
- Please re-read and check for typos. For example:
* "this paper intends to bridge", Introduction
Author response: Thank you for the suggestion. The authors have re-checked the manuscript and modified the typo mistakes.
Reviewer 2 Report
The paper entitled "ant colony optimization based feature engineering for improved ai models” studies similar framework using optimization algorithm. The author proposes a Ant Colony Optimization, incorporating diverse feature selection and extraction methods on text data. I found the article to be interesting and insightful. I think the authors have tried to put their findings into context.
The major points are the following:
1. The current title is misleading, I recommend to change it. The author aim is sentiment analysis, as explain in manuscript. Its better, if the author choose a title. In my opinion “Sentiment analysis or Textual feature extraction for Hate speech classification ”.
2. The citation in introduction 1st paragraph is require. Note: the citation must be most recent and related to the manuscript.
3. The contributions are not mentioned clearly. I suggest to add the main contributions as list. It would be easy for the reader of this journal to caught your contribution easily.
4. Rewrite and reorganized the literature (Table 1, page 3). The tabular form is much easy to get information but it also skip the important information, use paragraph with main ideas instead of table.
5. In Section 3.1, line 126, the author talk about “sentiment analysis”, and “hate speech”. But there is no information in introduction, and related work. Most importantly you should also added this “sentiment analysis”, and “hate speech” in your abstract.
6. Add a Table for data statistics, which represent Hate speech and stoke datasets in Section 3.1. Use Bar chart for Figure 1. Remove the Figure 2, here the world cloud is not needed.
7. Move Figure 3 to line 178, Figure 5 to 196. Futhermore, the caption of Figures and Tables must contained enough information.
8. Could you highlight what feature exactly the author extracted. Furthermore, give example in table.
9. The proposed architecture is unclear. Line 399 “different feature extraction”, what does the author mean here.
10. Table 3 the author provided accuracy, but in textual analysis its important to provide evaluation metrics (F1 score, precision and recall). Add these evaluation metrics too. Additionally the Figure 11 is unclear. Add Figures with high resolution .
Author Response
Dear Reviewer,
Thank you for the suggestions,
We have modified the manuscripts according to your comments.
Kindly consider.
- The current title is misleading, so I recommend changing it. The author's aim is sentiment analysis, as explained in the manuscript. It's better, if the author chooses a title. In my opinion “Sentiment analysis or Textual feature extraction for Hate speech classification ”.
Author Response 1: Thank you for the suggestion. The authors have changed the title.
“Textual feature extraction using Ant Colony Optimization for Hate speech classification”
- The citation in the introduction 1st paragraph is required. Note: the citation must be the most recent and related to the manuscript.
Author Response 2: Thank you for your suggestion. The authors have added citations for statements in the paper. The authors have added citations from [61] to [76].
[61] Linardatos, P.; Papastefanopoulos, V.; Kotsiantis, S. Explainable AI: A Review of Machine Learning Interpretability Methods. Entropy 2021, 23, 18. https://doi.org/10.3390/e23010018
[62] Ahmad, S.R., Bakar, A.A., & Yaakub, M.R. (2019). Ant colony optimization for text feature selection in sentiment analysis. Intell. Data Anal., 23, 133-158.
[63] Ting, T. O., Saraç, Esra, Özel, Selma AyÅŸe, 2014, "An Ant Colony Optimization Based Feature Selection for Web Page Classification", - https://doi.org/10.1155/2014/649260, The Scientific World Journal, Hindawi Publishing Corporation
[64] Fan C, Chen M, Wang X, Wang J and Huang B (2021) A Review on Data Preprocessing Techniques Toward Efficient and Reliable Knowledge Discovery From Building Operational Data. Front. Energy Res. 9:652801. doi: 10.3389/fenrg.2021.652801
[65] Kumar, Sachan Rohit and Kushwaha Dharmender Singh. “Nature-Inspired Optimization Algorithms: Research Direction and Survey.” (2021).
[66] Rodrigues, D., Yang, XS., de Souza, A.N., Papa, J.P. (2015). Binary Flower Pollination Algorithm and Its Application to Feature Selection. In: Yang, XS. (eds) Recent Advances in Swarm Intelligence and Evolutionary Computation. Studies in Computational Intelligence, vol 585. Springer, Cham. https://doi.org/10.1007/978-3-319-13826-8_5
[67] Hema Banati and Monika Bajaj, Fire Fly Based Feature Selection Approach, IJCSI International Journal of Computer Science Issues, Vol. 8, Issue 4, No 2, July 2011
[68] Amelio, A., Bonifazi, G., Corradini, E. et al. A Multilayer Network-Based Approach to Represent, Explore and Handle Convolutional Neural Networks. Cogn Comput (2022). https://doi.org/10.1007/s12559-022-10084-6
[69] Alessia Amelio, Gianluca Bonifazi, Francesco Cauteruccio, Enrico Corradini, Michele Marchetti, Domenico Ursino, Luca Virgili, Representation and compression of Residual Neural Networks through a multilayer network based approach, Expert Systems with Applications, Volume 215, 2023, 119391, ISSN 0957-4174, https://doi.org/10.1016/j.eswa.2022.119391.
[70] Lundberg, S. M., Nair, B., Vavilala, M. S., Horibe, M., Eisses, M. J., Adams, T., ... & Lee, S. I. (2018). Explainable machine-learning predictions for the prevention of hypoxaemia during surgery. Nature biomedical engineering, 2(10), 749-760.
[71] Najafabadi, M. M., Villanustre, F., Khoshgoftaar, T. M., Seliya, N., Wald, R., & Muharemagic, E. (2015). Deep learning applications and challenges in big data analytics. Journal of big data, 2(1), 1-21.
[72] Gao, S., & Ng, Y. K. (2022). Generating extractive sentiment summaries for natural language user queries on products. ACM SIGAPP Applied Computing Review, 22(2), 5-20.
[73] Gao, S., & Ng, Y. K. (2022). Generating extractive sentiment summaries for natural language user queries on products. ACM SIGAPP Applied Computing Review, 22(2), 5-20.
[74] Wei, Xianmin. (2014). Parameters Analysis for Basic Ant Colony Optimization Algorithm in TSP. International Journal of u- and e-Service, Science and Technology. 7. 159-170. 10.14257/ijunesst.2014.7.4.16.
[75] Shima Kashef, Hossein Nezamabadi-pour,An advanced ACO algorithm for feature subset selection,Neurocomputing,Volume 147,2015,Pages 271-279,ISSN 0925-2312,https://doi.org/10.1016/j.neucom.2014.06.067.
[76] H. S. Alghamdi, H. L. Tang and S. Alshomrani, "Hybrid ACO and TOFA feature selection approach for text classification," 2012 IEEE Congress on Evolutionary Computation, Brisbane, QLD, Australia, 2012, pp. 1-6, doi: 10.1109/CEC.2012.6252960.
- The contributions are not mentioned clearly. I suggest adding the main contributions as a list. It would be easy for the reader of this journal to catch your contribution easily. sg
Author Response 3: Thank you for your suggestion. The authors have added main contributions in the list format. Please refer to page number 3 highlighted with yellow color.
The authors’ investigation is summarized as follows:
- Implement the framework using Ant Colony Optimization (ACO) for feature
selection to improve the AI models.
- Evaluate these selected features on textual and numerical datasets.
- Also evaluate and analyze the Logistic Regression (LR), K-Nearest Neighbor (KNN), Stochastic Gradient Descent (SGD), and Random Forest (RF) machine learning algorithms.
- Rewrite and reorganize the literature (Table 1, page 3). The tabular form is much easier to get information from, but it also skips the important information, using paragraphs with main ideas instead of tables.
Author Response 4: thank you for the suggestion. The authors have added a description of table 1 on page number 4, highlighted with yellow color.
- In Section 3.1, line 126, the author talks about “sentiment analysis”, and “hate speech”. But there is no information in the introduction, and related work. Most importantly you should also add this “sentiment analysis”, and “hate speech” in your abstract.
Author Response 5: Thank you for the suggestion. The authors have added lines on page number 1 in the abstract and on page number 2 and 3 in the introduction section. Text highlighted with yellow color.
“The proposed feature selection and feature extraction techniques assist in enhancing the performance of the machine learning model. This research article considers numerical and text-based datasets for stroke prediction and detecting hate speech, respectively. The text dataset is prepared by extracting tweets consisting of positive, negative, and neutral sentiments from Twitter API.”
- Add a Table for data statistics, which represent Hate speech and stoke datasets in Section 3.1. Use Bar chart for Figure 1. Remove Figure 2, here the world cloud is not needed. dd
Author Response 6: thank you for the suggestion. Table number added on the page number representing the datasets. The authors, too, changed figure 1 to a bar chart for better visibility.
The authors feel that figure 2 is necessary as it improves the readability of the article. Hence it is retained. We hope the decision complies with the views of the reviewer.
Figure 1. Bar chart showing the distribution of Sentiment Labels in the text dataset.
Table 2: Dataset details shown with different characteristics.
Name of Dataset |
Number of samples |
Balanced/ Unbalanced |
Resampling Technique |
No. of labels |
Name of labels |
Stroke Prediction (Numerical Data) |
5110 Stroke:249 No stroke: 4861 |
Unbalanced |
SMOTE Oversampling |
2 |
Stroke and No stroke |
Twitter Data on Extremism (Text Data) |
93501 Positive (2): 29652 Negative (0): 33880 Neutral (1): 29969 |
Balanced |
None |
3 |
Positive, Negative and Neutral |
- Move Figure 3 to line 178, Figure 5 to 196. Furthermore, the caption of Figures and Tables must contain enough information.
Author Response 7: Thank you for the suggestion. The authors have moved figure 3 and figure 5 as per suggestions by reviewers. Captions of figures and tables are changed and made more informative. Captions of figures and tables highlighted with yellow color.
- Could you highlight what feature exactly the author extracted? Furthermore, give an example in the table.
Author Response 8: Thank you for the suggestion. The authors have mentioned line number 420 and 421. And also, in table 3, the authors have mentioned in table 3.
Table 3. Parameter settings of Ant Colony Optimization algorithm.
- The proposed architecture is unclear. Line 399 “different feature extraction”, what does the author mean here.
Author Response 9: Thank you for the suggestion. The authors have mentioned line number 420 and 421. And also, in table 3, the authors have mentioned in table 3.
Table 3. Parameter settings of Ant Colony Optimization algorithm.
Response:
- Table 3 the author provided accuracy, but in textual analysis it is important to provide evaluation metrics (F1 score, precision and recall). Add these evaluation metrics too. Additionally, Figure 11 is unclear. Add Figures with high resolution.
Author Response 10: Thank you for the suggestion. The authors have provided ROC-AUC scores along with accuracy as performance measures. A false positive rate curve is also provided in Figures 12 and 13. ROC score shows reciprocity between the true positive rate and the false positive rate. Table 5 shows both the accuracy and ROC-AUC curve. We hope the decision complies with the views of the reviewer. The resolution of Figure 11 changed.
Reviewer 3 Report
Please find attached the review results.

Author Response
Dear Reviewer,
Thank You for your Suggestions,
We have modified manuscript as per you suggestion,
Kindly consider.
(1) Literature review can be further discussed by focusing on the pros/cons of the existing methods in the literature.
Author Response 1: Thank you for the suggestion. The authors have added limitations of existing literature in table 1 on page number 4.
(2) Please provide additional information regarding ACO parameters. How are the values presented in the tables selected? Are they based on experience/simulation/rule of thumbs?
Author Response 2: Thank you for the suggestion. The authors have referred to literature from [1] to decide the parameters to be used. The authors also provided additional information in section 7.1, parameter setting.
[74] Wei, Xianmin. (2014). Parameters Analysis for Basic Ant Colony Optimization Algorithm in TSP. International Journal of u- and e-Service, Science and Technology. 7. 159-170. 10.14257/ijunesst.2014.7.4.16.
(3) Regarding ACO algorithm, there is not any reference to the representation format of an
individual/solution in the population. An illustrative example will be useful. In addition to this,
Which is the formula for the fitness function (a single AUC value or ACC value)? Since ACO is used to select subset of features each time and evaluate the training performance, the process should be reported as a wrapper approach.
Author Response 3: Thank you for the suggestion. The authors have added the following lines in the section reference mentioned in section 7.1. The authors have used the best accuracy for the fitness function to select the subset of features.
“A random forest classifier is incorporated within the Ant Colony Optimization algorithm that helps evaluate the fitness of each feature subset created by ants. In return, it gives the best fitness score, best accuracy, and selected features set. To assess the robustness of each set of selected features by each ant, we have employed a random forest classifier that returns the best fitness score, best accuracy, and the number of features selected.”
(4) Please check some representation errors in pages 12 and 13. This manuscript may be proposed for publication if it is addressed in the specified issues (accept with major changes).
Author Response 4: Thank you for the suggestion. The authors have checked representation errors on pages 12 and 13. And corrected them.
Reviewer 4 Report
The paper is focused on the use of the Ant Colony Optimization (ACO), a very traditional metaheuristic, in the context of feature engineering as a previous stage of machine learning models. Specifically, the authors claim that the main novelty of the paper is the use of feature engineering in text-based data. Even though this hypothesis at first could be interesting to explore, the paper needs to be improved in several directions for covering it in an appropriate way.
Main issues:
- The literature review section lacks of a comprehensive analysis that justifies the necessity of performing this research centered on the use of ACO for feature selection in textual data. This section is currently limited to present an overall table of the studied papers, without the adequate analysis.
-Section 6 proposes the use of ACO for feature selection in text and numerical datasets. Here it is necessary to point out that this topic have been extensively covered by the literature [1,2], and the authors do not properly justify, particularly in Figure 10, the novel issue they are current in the current work. The work lacks of novelty in this direction.
[1] Kashef, S., & Nezamabadi-pour, H. (2015). An advanced ACO algorithm for feature subset selection. Neurocomputing, 147, 271-279.
[2] Alghamdi, H. S., Tang, H. L., & Alshomrani, S. (2012, June). Hybrid ACO and TOFA feature selection approach for text classification. In 2012 IEEE Congress on Evolutionary Computation (pp. 1-6). IEEE.
Other structure problems:
-The evaluated or proposed models, presented across Sections 3-5, are very difficult to follow. In the beginning of Section 3, there are some references to datasets, that should be moved to the Experimental Results section. Furthermore, Section 3.2 makes reference to some data preprocessing techniques, like SMOTE, that are not directly connected to the goal of this work.
-Section 3.3, entitled Feature Engineering techniques, makes reference to feature extraction and selection approaches, but not directly related with ACO, that is the declared main focus of the paper, according to the title and abstract.
-Section 4 presents general machine learning models, that should be moved to a previous section focused on the background theory around the paper.
Considering these problems of novelty and structure, we think that the paper should be deeply transformed to be considered as possible publication in the journal. At this stage we should suggest Paper Rejection.
Author Response
Dear Reviewer,
Thank You for your Suggestions,
We have modified manuscript as per you suggestion,
Kindly consider.
- The literature review section lacks a comprehensive analysis that justifies the necessity of performing this research centered on the use of ACO for feature selection in textual data. This section is currently limited to present an overall table of the studied papers, without the adequate analysis.
Author Response 1: Thank you for the suggestion. The authors have added a detailed analysis of table 1 in the literature review section on page number 4. The authors have added a description of table 1 on page number 4, highlighted with yellow color.
2) Section 6 proposes the use of ACO for feature selection in text and numerical datasets. Here it is necessary to point out that this topic has been extensively covered by the literature [1,2], and the authors do not properly justify, particularly in Figure 10, the novel issue they are current in the current work. The work lacks novelty in this direction.
[1] Kashef, S., & Nezamabadi-pour, H. (2015). An advanced ACO algorithm for feature subset selection. Neurocomputing, 147, 271-279.
[2] Alghamdi, H. S., Tang, H. L., & Alshomrani, S. (2012, June). Hybrid ACO and TOFA feature selection approach for text classification. In 2012 IEEE Congress on Evolutionary Computation (pp. 1-6). IEEE.
Author Response 2: Thank you for the suggestion. The authors have studied the articles suggested by reviewers and included them, as these articles have extensively covered ACO for feature subset extraction. The authors have mentioned novelty in section 1.1, Motivation behind the proposed work. And some lines also have been added on page number 14 in section 6 proposed architecture.
3) Other structure problems:
The evaluated or proposed models, presented across Sections 3-5, are very difficult to follow. In the beginning of Section 3, there are some references to datasets that should be moved to the Experimental Results section. Furthermore, Section 3.2 makes reference to some data preprocessing techniques, like SMOTE, that are not directly connected to the goal of this work.
Author Response 3: Thank you for the suggestion. The authors have removed the reference to datasets from section 3 and added them to the experimental results. Similarly, the reference from section 3.2 has been removed as per the reviewers’ comments.
3) Section 3.3, entitled Feature Engineering techniques, makes reference to feature extraction and selection approaches, but not directly related with ACO, that is the declared main focus of the paper, according to the title and abstract.
Author Response 3: Thank you for the suggestion. As per the proposed method, the authors have presented the textual feature extraction approaches in section 3.3.1. similarly, in section 3.3.2, feature selection approaches are presented. The authors also explained the need for ant colony optimization in feature selection on page number 13 in section 5. Also, in proposed methodology presented the flow of architecture, which includes steps such as feature extraction and feature selection using ant colony architecture in figure 9 Ant Colony Optimization Architecture for feature selection.
Table 5 shows the Performance Comparison of the best models for the Textual and Numerical Dataset that evaluated the methodology used for feature extraction and ACO.
4) Section 4 presents general machine learning models that should be moved to a previous section focused on the background theory around the paper.
Author Response 4: Thank you for the suggestion. The authors have explained general machine-learning algorithms in the context of the whole proposed methodology, as shown in figure 10, so it is placed after feature selection. We hope the decision complies with the views of the reviewer.
Round 2
Reviewer 3 Report
Authors put a special effort on improving the manuscript according suggestions. As a result, I recommend the manuscript for publications in the journal.
Reviewer 4 Report
The authors have done a great effort to improve the paper according to my previous comments. The current version of the manuscript is clearer and appropriate for publication.
I suggest acceptance.